# Effects of Bit Chewing on Gastric Emptying, Small Intestinal Transit, and Orocecal Transit Times in Clinically Normal Horses

**DOI:** 10.3390/ani13152518

**Published:** 2023-08-04

**Authors:** Molly E. Patton, Frank M. Andrews, Sophie H. Bogers, David Wong, Harold C. McKenzie, Stephen R. Werre, Christopher R. Byron

**Affiliations:** 1Department of Large Animal Clinical Sciences, Virginia-Maryland College of Veterinary Medicine, Virginia Tech, 205 Duck Pond Drive, Blacksburg, VA 24061, USA; mepatton@vt.edu (M.E.P.); s0phie@vt.edu (S.H.B.); hmckenzi@vt.edu (H.C.M.III); 2Department of Veterinary Clinical Sciences, School of Veterinary Medicine, Louisiana State University, Baton Rouge, LA 70803, USA; fandrews@lsu.edu; 3Department of Veterinary Clinical Sciences, College of Veterinary Medicine, Iowa State University, 1800 Christensen Drive, Ames, IW 50011, USA; dwong@iastate.edu; 4Laboratory for Study Design and Statistical Analysis, Virginia-Maryland College of Veterinary Medicine, Virginia Tech, 205 Duck Pond Drive, Blacksburg, VA 24061, USA; swerre@vt.edu

**Keywords:** horse, ileus, colic, bit chewing, gastrointestinal motility

## Abstract

**Simple Summary:**

Ileus, or a temporary lack of intestinal motility, is a common life-threatening problem in horses, especially following abdominal surgery. Current treatments have variable efficacy and a high cost. In human patients suffering from ileus, sham feeding in the form of gum chewing has shown promising results in improving clinical signs and increasing gastrointestinal motility. Bit chewing, a form of sham feeding for horses, has also been proven to decrease gastrointestinal (GI) total transit time (TTT); however, ileus in horses typically affects the small intestine, a part of the GI tract that has not been investigated in regard to bit chewing. Our objective was to determine whether bit chewing shortens the gastric emptying time (GET), small intestinal transit time (SITT), and total orocecal transit time (OCTT) in clinically normal horses. Gastrointestinal motility in horses was compared between bit-chewing conditions and control (no bit chewing) conditions in a prospective crossover design study using acetaminophen serum samples as a marker for GET and video endoscopy (ALICAM) capsules to determine GET, SITT, and OCTT. The results indicate no adverse effects and significantly shortened OCTT after bit chewing. Bit chewing is potentially a safe, inexpensive, and effective clinical treatment to improve small intestinal motility in horses.

**Abstract:**

Ileus is a common life-threatening problem in horses, and currently available treatments may be ineffective. The purpose of this study was to determine whether bit chewing, a form of sham feeding, decreases the gastric emptying time (GET), small intestinal transit time (SITT), and total orocecal transit time (OCTT) in clinically normal horses in a prospective crossover study. Nine healthy horses were acclimated and fed a standardized diet. Following 24 h of fasting, self-contained video endoscopy capsules and acetaminophen were administered into the stomach via a nasogastric tube. Each horse underwent experimental (bit chewing for 20 min every 6 h) or control (no bit chewing) conditions, with a 3-week minimum washout period between conditions. The horses were enrolled in either part of the study until all video capsules were retrieved and/or 30 days lapsed. The video capsules were recovered from manure, and GET, SITT, and OCTT were determined from a video analysis. Bit chewing significantly decreased OCTT (*p* = 0.015) compared to the control conditions. Bit chewing decreased GET and SITT, but the differences were not significant. The mean (median) times determined via the video capsule analysis for the bit-chewing conditions were as follows: GET, 2.34 h (2.86 h); SITT, 3.22 h (3.65 h); and OCTT, 5.13 h (6.15 h), and for the control conditions, they were as follows: GET, 3.93 h (5 h); SITT, 3.79 h (4.4 h); and OCTT, 8.02 h (9.92 h). Bit chewing decreased OCTT in healthy horses. Because this segment of the gastrointestinal tract is frequently affected by ileus, bit chewing may be a safe and inexpensive intervention for that condition in horses. Further investigation in clinical patients with ileus is warranted.

## 1. Introduction

Post-operative ileus (POI) is a recognized complication following equine abdominal surgery. The reported prevalence varies from 18.4 to 33% [1,2,3,4,5]. This percentage increases significantly if a small intestinal lesion is present [1,3,5]. The complications associated with POI include an increased likelihood of repeat laparotomy, an increased cost to owners, and prolonged hospitalization [6,7]. The mortality rate of horses that develop POI has been reported to be as high as 86% [7]. The pathogenesis of POI is not completely understood, but it is thought to be a result of neurogenic, inflammatory, and pharmacological insults to the gastrointestinal (GI) tract [7]. There is currently no specific treatment that is considered completely effective in preventing or treating equine POI. For this reason, multimodal treatment approaches, such as prompt surgical intervention with a good surgical technique, gastric decompression, judicious medication administration, and intravenous fluid therapy, are employed with varying success [7,8,9]. In humans, several additional nonpharmacological treatment methods are also added to a patient’s treatment regimen, including early ambulation and early feeding, if tolerated [10]. Sham feeding, most commonly in the form of gum chewing, is used in human hospitals as an inexpensive and simple method to stimulate the cephalic–vagal response without overwhelming the GI tract with bulk food before it is tolerated. Previous reports have speculated that gum chewing is a safe, inexpensive, simple, and well tolerated way to successfully reduce the time from surgery to the first fecal passage [11,12,13,14] and reduce hospitalization time in people [11,13].

The idea of sham feeding in the form of bit chewing for horses was first evaluated using snaffle bits placed in the mouth of ten adult horses, and short (<3 s) borborygmi sounds were found to be significantly increased in the first five minutes of bit application [15]. Bit chewing was well tolerated by all horses in that study, and no negative side effects were reported. Other investigators [16] further evaluated bit chewing in the form of sham feeding in a prospective crossover design study of six adult horses. The results indicated that the horses had a significantly shorter total transit time of non-digestible markers when bit chewing was implemented as compared to control (no bit) conditions [16]. While both of these studies provide promising information regarding the effects of bit chewing on GI motility in horses, the effects of bit chewing on specific intestinal segments remain unknown. Specifically, the effects of bit chewing on SI transit time is of important value, as this is the part of the GI tract that is most commonly affected by ileus in horses.

The purpose of this study was to evaluate the effects of bit chewing as a form of sham feeding on the gastric emptying time (GET), small intestinal transit time (SITT), and overall orocecal transit time (OCTT) in clinically normal horses. The hypothesis was that bit chewing shortens GET, SITT, and OCTT compared with control (no bit) conditions. The goal of this study was to provide evidence that bit chewing may potentially be used as an inexpensive, simple, and prokinetic tool in horses suffering from POI and other causes of ileus.

## 2. Materials and Methods

### 2.1. Animals

All procedures were approved by the Institutional Animal Care and Use Committee of Virginia Tech (Protocol #20–173). One horse (15-year-old Paint Mare, 497 kg) was enrolled in a pilot study to evaluate video capsule image quality and the gastrointestinal passage prior to study commencement. In order to minimize the number of horses needed, a crossover design was used, allowing each animal to serve as its own control. Nine healthy horses in total were included in the study, not including the mare in the pilot study. The breeds used in the study included Quarter Horse (*n* = 6), Appaloosa (*n* = 1), Tennessee Walking Horse (*n* = 1), and Warmblood (*n* = 1). There were 3 geldings and 6 mares ranging in age from 7 to 23 years (median 15 years), and their body weights ranged from 401 to 559 kg (median 527 kg). The mare in the pilot study was not included in the experimental study. Three horses were enrolled in the study after three of the six initial horses did not complete the study requirements (i.e., did not pass one or more camera capsules) in a one-month period and were subsequently removed. The results of the acetaminophen absorption times of the three horses that were removed were included in the study. All horses used in the study were client-owned, with written consent obtained prior to enrollment. The inclusion criteria included having no history of GI disease, normal results of physical examination, and being fed a consistent diet (grass hay and fresh grass only) for at least one week prior to the start of the study. Three horses were euthanized following study completion at the owners’ request due to reasons unrelated to the study. The reasons for euthanasia included chronic lameness, melanomas, and behavior. None of the horses that were excluded from the study were the horses that were euthanized.

### 2.2. Study Design

The study was conducted as a prospective 2-period crossover allowing each animal to serve as its own control. The horses were evaluated with each treatment (bit or no bit (control)) for two weeks and had a minimum 4-week washout period between treatments. The horses were housed in temperature-controlled barns to reduce the influence of changing weather patterns. For each group, horses on identical feed were brought in from the same pasture and placed in individual 3 × 3 m stalls in an environmentally controlled barn, where the horses could see other horses in the stalls. The horses underwent a 24 h acclimation period prior to study initiation. The horses were randomly assigned to a bit-chewing group or control (no bit) group via a coin toss for the first trial period. After the first trial period, all horses were returned back to the same pasture for the washout period. The second trial period was conducted in the exact same fashion, with the horses assigned to the other group (bit chewing or control).

### 2.3. Feeding Schedule

To standardize the feeding schedule and simulate conditions similar to those of a horse undergoing treatment for colic, the horses were fasted for twelve hours prior to the start of the study (time = 0). An additional 12 h fast following video capsule and acetaminophen (used to evaluate gastric emptying) administration was implemented to maximize the visualization of the capsule images (time 0–12). Following this fast, a gradual and consistent refeeding schedule (Table 1) was implemented in each horse to standardize the total feed intake throughout the entire project. This feeding plan was similar to a refeeding schedule for a horse recovering from abdominal exploratory surgery. A gradual increase in pelleted food was provided from hour 12 until hour 48 of the study (500 g of pellets every 4 h). Grass hay was re-introduced starting from hour 25 to 48 (200–250 g of hay every 4 h from hour 24 to 36, then 500–600 g of hay every 4 h from hour 36 to 48) and was increased based on 2% total body intake for each horse’s weight over a period of 4 days (25% of total body intake from hour 48 to 72, 50% total body intake from hour 72 to 96, 75% total body intake from hour 96 to 120, and then 100% total body intake for the remainder of the study). These feedings were divided into small, frequent (every 4 h) meals throughout a 24 h period. After Day 14, if a horse had not passed a capsule and remained in the study, the horse was fed free-choice hay until the capsules were obtained from the feces or an additional two weeks lapsed.

### 2.4. Transit Markers

To measure GET, SITT, and OCTT, 3 video endoscopy capsules (ALICAM, Infiniti Medical, Palo Alto, CA, USA) were administered to each horse without sedation via a nasogastric (NG) tube as previously described [17] after the initial 12 h fast. Three capsules were used per horse to increase the success rate of collecting the capsules in the manure. Following capsule administration, acetaminophen (20 mg/kg) was administered as a slurry through the NG tube as another method of evaluating gastric emptying as previously described [18]. The NG tube was then removed, and each horse was fasted for an additional 12 h.

### 2.5. Treatment: Bit Chewing

Bit placement was initiated at time 0 after the initial 12-h fasting period, immediately following video capsule and acetaminophen administration via nasogastric intubation. Each horse in the bit-chewing group had an apple-flavored snaffle bit (Horze Equestrian, Watertown, CT, USA) placed in their mouth [16] for twenty minutes every six hours until all capsules were retrieved or until the end of the study period (i.e., one month, prior to the washout period), whichever came first. A twenty-minute time period was selected, as it was previously determined that the application of a bit led to consistent salivation and mastication for a least a twenty-minute period [15].

The bit was applied and kept in place using a modified head piece fixed at a location in the interdental space that led to consistent mastication. Saliva production and swallowing for the entire 20 min of the bit placement were observed in all horses. If chewing decreased during the 20 min study period, a small amount of molasses was placed onto the bit to successfully encourage chewing. A 12cc syringe was filled with molasses to drizzle it onto the bit. One investigator (M.E.P.) observed each horse for the entire bit-chewing period to ensure adequate mastication. Adequate bit chewing was defined as constantly chewing (at least one chew every 1–2 s) for the 20 min duration. Bit chewing was not performed at the same time as feeding to eliminate cephalic–vagal stimulation with food administration [19]. Food was withheld for one hour following bit chewing.

### 2.6. Acetaminophen: Gastric Emptying Time

An intravenous catheter was aseptically placed in the right or left jugular vein prior to NG intubation, and a baseline venous blood sample (5 mL) was obtained and placed in a heparinized blood collection tube (BD Vacutainer^®^, Franklin Lakes, NJ, USA). Following acetaminophen (20 mg/kg) administration via NG intubation, venous blood samples were collected at 15, 30, 45, 60, 75, 90, 105, and 120 min for an acetaminophen concentration analysis [18]. The intravenous catheter was removed after the final blood collection. The blood samples were immediately centrifuged, and plasma was stored at −80 °C until the acetaminophen concentration analysis could be performed. Plasma concentrations of N-acetyl-para-aminophenol (acetaminophen) were measured using liquid chromatography (Acquity H-Class UPLC and Xevo TQD mass spectrometer, Waters Corporation, Milford, MA, USA). The time to the maximum acetaminophen serum concentration and the area under the serum concentration vs. time curve (AUC) were determined using commercially available software (Phoenix WinNonlin 8.2, Pharsight Corporation, Princeton, NJ, USA), as performed in previous studies [20,21].

### 2.7. Video Endoscopy Capsules: Gastric Emptying Time, Small Intestinal Transit Time, and Orocecal Transit Time

All fecal material was collected from each horse’s individual stall every 3 h following video capsule administration using a shovel to minimize the chance for a capsule to fall and remain in the stall. The majority of the capsules were well embedded within the manure. Capsules within the fecal material were collected and counted via the manual sifting of manure or radiography every 3 h until all 3 capsules were passed, or 14 days had elapsed, whichever came first. If no capsules were retrieved after 14 days, the study period was extended until at least one of the three capsules was obtained, or an additional 14 days had passed (whichever came first). Following a total of one month of hospitalization, the horses were discharged from the hospital and returned to their owners to allow for a washout period if any capsules had been retrieved in the first trial period, or they were removed from the study if no capsules had been obtained. Three horses did not pass any capsule in the first trial period and were thus removed from the study; three new horses were enrolled, and the study was repeated.

When administered for the determination of GET, SITT, and OCTT, the transit times recorded from retrieved capsule time-stamped images were used to compare the transit times between treatment groups, as previously reported [17]. The serial number of each video capsule in each study period was recorded to ensure the accurate identification of the capsules. Following the collection of the video capsules, the video recordings from each capsule were downloaded and randomized (Excel Randomization). The time points when a capsule moved from the stomach to the duodenum (GET) and from the duodenum to the cecum (SITT), and the overall time from the stomach to the cecum (OCTT) were recorded for each capsule video by three blinded researchers (M.E.P., C.R.B., S.H.B.). The observers were trained to identify each particular segment of the GIT using previously published images of the stomach, small intestine, and cecum mucosa.

### 2.8. Statistical Analysis

A power calculation using an expected standard deviation difference of 0.5 [15] with a minimum detectable effect of 25%, an alpha of 0.05, and a power of 80% indicated that 6 horses using a crossover design study would be needed to detect significant differences between groups (Graph pad StatMate version 2.0, San Diego, CA, USA). Images from the ALICAM capsules were downloaded and analyzed using specialized ALICAM capsule software (ALICAM reader version SV3, Palo Alto, CA, USA). For horses with 2–3 capsules available, the median of the transit times (for each metric) was obtained (across capsules and observers) and used for downstream analyses. Normal probability plots were inspected to assess the distribution properties of the data. The time points from each blinded observer were compared as coefficient variants using Cohen’s kappa coefficient to assess interobserver variability. GET (acetaminophen and capsule data), SITT, and OCTT were compared using Wilcoxon signed-rank tests, and they are presented as median (interquartile range) [minimum–maximum], as the data were non-parametric and abnormally distributed. Data analyses were performed using SAS 9.4 software (SAS Institute Inc., Cary, NC, USA). The results were considered significant when *p* < 0.05.

## 3. Results

Bit chewing was tolerated well by all horses in this study without any complications or side effects. In one horse, NG intubation was not possible without administering xylazine (0.3 mg/kg, IV) due to temperament. The same xylazine dose was administered to that horse during capsule administration, and the timing of xylazine administration to nasogastric intubation (3 min) was the same in the second trial period to maintain consistency. The nasogastric tube was placed in the remainder of the horses without complication or difficulty. No complications associated with intravenous catheterization were observed for the entire study period. Acetaminophen administration was also well tolerated with no systemic side effects, and gastric emptying could successfully be determined using liquid chromatography.

Twenty-seven of the forty-five capsules that were administered to the horses were able to be successfully recovered from feces (*n* = 24) or in a post-mortem examination (*n* = 3). One of the twenty-seven capsules (control capsule #1) did not move from the stomach for the entire battery life of the capsule (Figure 1). Twelve of the twenty-seven capsules (bit-chewing capsule #8–10,12,13,15; control capsule #4,5,7,10,11) did not pass through the ileocecal orifice during the battery life of the capsule. The battery life of the endoscopy capsules ranged from 7.8 to 22.7 h. Of the video endoscopy capsules that were recovered, the time to recovery (mean, median, and range) in the manure was as follows: 11.14 days, 6.45 days, and 2.25–43 days. No video endoscopy capsules were recovered within 21 days from three horses; these horses were excluded, and the study was repeated with three different horses. The capsules were not difficult to use, and the transitions from the stomach to the duodenum and from the ileum to the cecum were easily identifiable with fair-to-excellent interobserver variability between the three blinded observers (Figure 2, Figure 3 and Figure 4). The coefficient variants were excellent for GET (CV = 0.98, *n* = 26 capsules, six horses), fair for SITT (CV = 0.36, *n* = 14 capsules, six horses), and good for OCTT (CV = 0.79, *n* = 14 capsules, six horses).

There was no difference in GET between the treatment and control horses as measured via acetaminophen liquid chromatography (*p* = 0.5; Table 2). The horses under the bit-chewing conditions had a reduced median GET of 0.5 h (0.25–0.75 h) compared to a median GET of 0.625 h (0.5–1 h) for the control conditions (*p* = 0.5) as determined via the acetaminophen analysis.

As determined via the video capsule analysis, the horses under the bit-chewing conditions had shorter GET (*p* = 0.14) and SITT (*p* = 0.89), and significantly shorter overall OCTT (*p* = 0.015) than under the control (no bit) conditions. The median (least squares mean) times for the bit-chewing conditions were as follows: GET, 2.86 h (2.34 h); SITT, 3.65 h (3.22 h); and OCTT, 6.15 h (5.13 h), whereas the median (least squares mean) times for the control conditions were as follows: GET, 5 h (3.93 h); SITT, 3.79 h (4.4 h); and OCTT, 9.92 h (8.02 h).

## 4. Discussion

The goal of this study was to determine the effects of bit chewing on the gastric and small intestinal motility of normal horses. Our results show that bit chewing significantly shortened OCTT compared to the control conditions. Although the GET and SITT differences were not statistically significant, both median values were shorter when the bit-chewing times were compared to the control. These findings may support the use of bit chewing as a simple and inexpensive method to augment GI motility in the horse.

POI remains a common problem in equine colic patients, especially in those with small intestinal disease. The current mainstay of treatment includes maintaining hydration and electrolyte balance with intravenous fluids, gastric decompression, and prokinetic therapy, as well as addressing any additional concerns, such as endotoxemia or laminitis. The prokinetic drugs that are commonly implemented include lidocaine, metoclopramide, bethanechol, and erythromycin [5,22,23,24,25]. Each of these drugs, as with other prokinetic pharmacologics, have potentially detrimental side effects and increase the cost of hospitalization [6]. Additionally, prokinetic medications have not shown complete efficacy in resolving POI. Due to these reasons, multimodal prokinetic therapy should be considered in the post-operative recovery plan. Established human post-operative protocols to decrease the incidence of POI include early ambulation, early feeding (if possible), and sham feeding in the form of chewing gum if full feed is not possible [10]. Sham feeding by chewing on a bit in equine patients can be an easy and inexpensive prokinetic method with no side effects.

The use of bit chewing as a form of sham feeding in horses was inspired from gum chewing in human medicine and its effect on increasing the cephalic–vagal response [13,26], but it has only limited clinical use to date [15,16]. The most recent publication investigating bit chewing in horses evaluated changes in the total gastrointestinal transit time; the results indicated that bit chewing significantly shortened the total transit time in clinically normal horses [16]. While this information is encouraging, an evaluation of the effects of bit chewing on the proximal GIT is of clinical importance, as post-operative ileus primarily involves the small intestine in the horse [6].

In this study, two relatively noninvasive methods to evaluate GI motility were used, including video endoscopy capsule images and acetaminophen absorption time. Acetaminophen is poorly absorbed in the stomach but is rapidly absorbed once it reaches the small intestine and thus can be effectively used to measure GET by taking serial blood samples [18]. This was first used in the horse in 1998 [18], where acetaminophen proved to be a practical, minimally invasive, and easy way to measure liquid-phase gastric emptying in horses, and it has since been successfully used in other studies to compare different medications and their effects on GET [27]. Video endoscopy capsules were also used to measure GET as well as SITT and overall OCTT in the present study. Steinmann and colleagues first described the use of these wireless endoscopy capsules in horses in 2020, with the proposed benefits compared to those of other capsule systems, including having four cameras with high-resolution LED lights, having the ability to record 360-degree diagnostic images, having the technology be completely ambulatory (therefore not needing any external memory device), and having a power-save mode when the capsule is stationary to prolong battery life [17]. The same benefits were observed in this study.

In this study, while OCTT was shortened in the bit-chewing group, significant differences in GET and SITT were not seen between treatments, although both GET and SITT, especially GET, trended shorter in the bit-chewing group than in the control. The possible reasons for the lack of significant results include the capsules becoming lodged in the stomach, making some capsule gastric emptying times delayed compared to other capsule and acetaminophen GETs from the same horse, as well as decreasing the amount of time in the small intestine before the capsule battery expired, making the SITT unable to be assessed. Increasing the sample size and finding a method for increasing battery life are methods that could be implemented to attempt to eliminate this problem in subsequent studies.

The major limitations reported in a previous study using ALICAM endoscopy capsules include the obstruction of images by feed material, a widely varied capsule excretion time and battery life, and the inability to confidently localize lesions, potentially limiting the use of capsules in a clinical setting [17]. The widely varied excretion time of the endoscopy capsules as seen by Steinmann et al. was also a limiting factor in this study [17]. Additionally, the physical characteristics of the capsules likely resulted in the substantially different GET versus acetaminophen, as indicated by our results. A recovery of only 24/45 capsules pre-mortem with a maximum excretion time of 43 days considerably increased the time and cost of the study due to the inaccessible data from the non-excreted capsules and requiring an increase in the number of horses to ultimately be enrolled in this study. This resulted in a greater number of horses for which there were acetaminophen gastric clearance data (*n* = 9) versus capsule transit data (*n* = 6). Additionally, the capsule battery life also varied greatly in this study, ranging from 7.8 to 22.7 h. A shortened battery life limited the capsules from reaching the cecum, and it made capsule data eligible to be used in evaluating GET only. This study also found a large discrepancy in stomach exit times for the capsules, and the capsules recovered from post-mortem examinations had settled in the ventral large colon or apex of the cecum. Although this may be an incidental finding, ALICAM capsules were designed for use in small animals. Therefore, it is unknown whether the volume of ingesta and the anatomy of the equine GIT require a specifically designed capsule to exit the stomach, cecum, and large colon. Potential design features that could be altered to increase performance in the GI tract are to alter the capsule buoyancy and decrease the capsule size to a size similar to a small piece of normal ingested food, such as grain. These limitations should be considered as potential complications when attempting to use wireless endoscopy capsules in research or clinical settings.

While the findings from this study provide further promising evidence that bit chewing may have a positive prokinetic effect on OCTT in clinically normal horses, this finding alone does not prove that bit chewing will increase GI motility in all normal horses or clinical horses suffering from ileus, and further studies to evaluate GET, SITT, and OCTT are warranted. Six horses in a crossover design study were deemed sufficient in our power analysis, but a larger sample size is preferred to prevent type II statistical errors. A future large, multi-center, randomized study evaluating the effects of bit chewing on POI would be beneficial. Overall, despite these limitations, OCTT was significantly hastened, suggesting that bit chewing could have a prokinetic effect on GI motility under certain conditions in normal horses, especially in conjunction with the acetaminophen GET data.

In conclusion, this study provides evidence that bit chewing is safe and well tolerated and leads to a significantly shorter OCTT versus control (no bit) conditions in clinically normal horses. Bit chewing is an inexpensive therapy to implement, and there were no side effects. This treatment could be used in conjunction with current POI treatment methods with little additional work or training for personnel. Additional investigations of the effects of bit chewing on clinical horses with clinical problems causing impaired GI motility are warranted.

## 5. Conclusions

Ileus is a common perioperative complication in horses with colic. While multiple pharmacologic treatment options have been investigated with varying degrees of success, each drug has potential side effects, variable efficacy, and an increased cost to owners. Other prokinetic therapies, such as sham feeding in the form of bit chewing, may provide an inexpensive method to stimulate the cephalic–vagal response with minimal additional time in treatment administration and no side effects. While not all data comparisons in the present study were statistically significant (gastric emptying and small intestinal transit times), the overall orocecal transit time was significantly shorter in horses during bit chewing than in control (no bit chewing) conditions. The limitations include a small sample size, the inability to obtain all capsules at the end of the study period, and a widely varying battery life among the capsules. Additional studies evaluating the specific effects of bit chewing on OCTT are warranted, ideally using multiple GI motility measurement modalities in horses with clinical disease.

## Figures and Tables

**Figure 1 animals-13-02518-f001:**
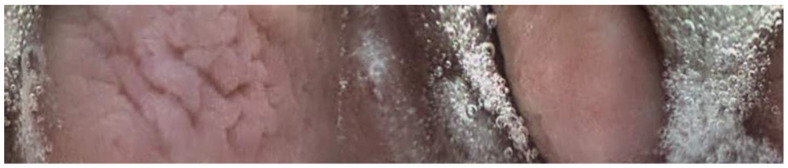
Picture obtained from endoscopy capsule of the gastric mucosa.

**Figure 2 animals-13-02518-f002:**
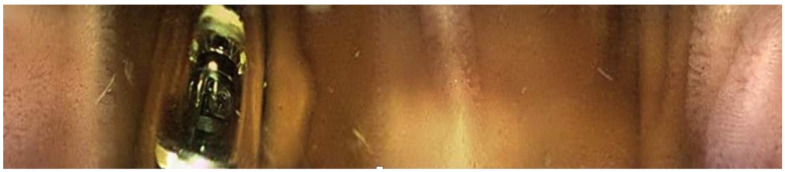
Picture obtained from endoscopy capsule within the small intestine. Note the “carpet-like”-appearing villi present within the small intestine compared to the more corrugated appearance of the stomach. An additional endoscopy capsule is also in the picture.

**Figure 3 animals-13-02518-f003:**
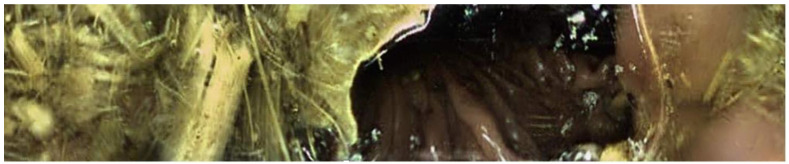
Picture obtained from endoscopy capsule at the level of the ileocecal orifice. Note the gas present within the base of the cecum.

**Figure 4 animals-13-02518-f004:**
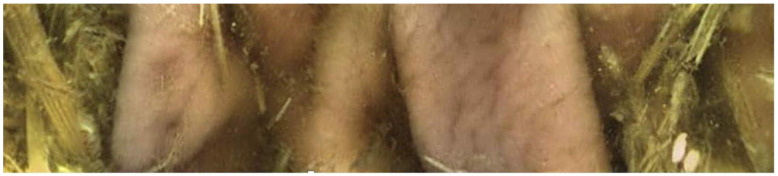
Picture obtained from endoscopy capsule within the cecum. Villi are not as obvious as in the small intestine, and the mucosa again has a more corrugated appearance, slightly paler than the stomach mucosa.

**Table 1 animals-13-02518-t001:** Feeding schedule of horses enrolled in study. Total body intake of hay measured at 2% of each horse’s body weight (g = grams).

Day −1	Hour −12 to 0: Initial 12 h fast
Day 0	Nasogastric intubation, additional 12 h fast Hour 12–24: 500 g pelleted feed every 4 h
Day 1	Hour 25–36: 500 g pelleted feed + 200–250 g grass hay every 4 h Hour 36–48: 500 g pelleted feed + 500–600 g grass hay every 4 h
Day 2	25% total body intake grass hay divided into feedings every 4 h
Day 3	50% total body intake grass hay divided into feedings every 4 h
Day 4	75% total body intake grass hay divided into feedings every 4 h
Day 5	100% total body intake grass hay divided into feedings every 4 h
Day 14+	Free-choice hay

**Table 2 animals-13-02518-t002:** Summary of GET, SITT, and OCTT of the study population when bit chewing compared to the control. Numbers are expressed in hours as a median time (interquartile range) [minimum–maximum]. Statistically significant differences (*p* < 0.05) are bolded with an asterisk (*).

	Gastric Emptying Time (Acetaminophen)	Gastric Emptying Time (Endoscopy Capsule)	Small Intestinal Transit Time (Endoscopy Capsule)	Orocecal Transit Time (Endoscopy Capsule)
*Bit Chewing*	0.5 h (0.5) [0.25–0.75 h]	2.86 h (5.07) [0.36–12 h]	3.65 h (0.88) [2.22–7.2 h]	6.15 h (0.88) [4.68–8.94 h]
*Control*	0.625 h (0.50) [0.5–1 h]	5 h (4.73) [1.76–15.4 h]	3.79 h (1.68) [2.88–8.9 h]	9.92 h (1.77) [8.73–10.97 h]
*p-value*	0.50	0.14	0.89	0.015 *

## Data Availability

The data presented in this study are available in the article.

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
