# Peer review of "Effects of Bit Chewing on Gastric Emptying, Small Intestinal Transit, and Orocecal Transit Times in Clinically Normal Horses"

_animals, 2023, doi:10.3390/ani13152518_

Round 1
Reviewer 1 Report
REVIEW REPORT
Animals-2491040-peer-review
This original research paper looks at the hypothesis that bit chewing shortens gastric emptying time, small intestinal transit time, and overall orocecal transit time compared with bitless self-controls in a crossover study design. The authors evaluated horses for a two-week period with 4 weeks washout, in a homogenous setting.
· The paper’s premise is original (assessing GET, SITT, OTT as influenced by bit chewing as a form of sham eating) and well-defined as one hypothesis. Some of the measured parameters have been documented before in horses, however, more research is needed.
· The work fits the journal scope, as this research might have implications regarding equines of all uses and improves knowledge of the (patho)physiology of the GI tract.
· The abstract is relevant to the paper, and the title fits the design of this study.
· The sample size was small. Power calculations supported the use of six horses in this study, however, statistical significance was weak for GET and SITT changes, with a trend towards shortened GET especially- this might improve with a larger sample size.
· The paper is written correctly, the supporting figures are suggestive and understandable and help interpret the presented data. Methods were thorough and well described, and fit the purpose of the paper. The data is however not very robust for final conclusion drawing as the sample size was small.
· The conclusions of the study are relevant to the field, as POI prevention is key in ensuring better surgery outcomes in equines, and a multimodal approach is clearly needed- the principle of “every little bit counts” might work here as well.
· Please revise references as there are some misnumberings in the body (reference number does not fit with the information cited)- see line 151, 168, 183 and all the rest afterwards. Please check line 397 and 398- there is an error in numbering citations and this has misnumbered the rest of the citation in text as described above.
· This paper does present an interesting option regarding intestinal transit time manipulation and would have benefitted from a larger sample size.
Other observations regarding minor grammatical errors/ changes:
line 225- replace tub with tube
Reviewer 2 Report
The authors provide a clearly written evaluation of a practical treatment (bit chewing) in clinically healthy horses. This makes a valuable contribution to the literature on a relatively novel intervention. There are minor typos and area for improved clarity particularly in the methods and statistical handling of the data. Citations frequently are not appropriate (I suspect they were shifted at some point in the editing) and should be carefully reviewed before resubmission including content of references and the text in the selected references.
Specific comments from text:
Abstract line 43-44: Specify the size and P value associated with this decrease in OCTT
Introduction Line 55: This reference does not determine these numbers directly please cite primary sources for prevalence numbers.
Methods 135-138: Were feedings every 4 hours or how were they divided on day 2-14?
Methods: Unclear how the 3 extra horses data was handled. Presumably acetaminophen gastric emptying time was available in those 3 horses? Were they excluded all together? Were any of these horses the horses that were subsequently euthanized?
Line 170-183: Clarify how GET was calculated from Acetaminophen AUC and Max serum concentration. This is not calculated in indicated references which are also incorrectly listed.
References: References seem off? 13 should be 14 etc.
Statistics Line 209 : minimum detectable effect should be specified for this power calculation.
Statistics 209-219: How was normality assessed?
Results: 223 Was timing for xylazine the same in both trial periods? Please edit for clarity.
Line 226- was should read were observed
Results - Please include a measure of variability (IQR or STD) based on normality in addition to range data as it is not possible to determine if data are clustered based on the provided range.
Was the correlation between GET by acetominophen and GET by capsule evaluated? If not please consider and if so please include in the results.
Reference Line 397-398 (Lisowski reference is listed as 2. And 3.
Figures – Figure 1 not referenced in the text, may be best presented as Figure 1 (a-d)
Overall this is a valuable contribution to the literature that will benefit from statistical consideration and general editing.
Minor typos were identified but overall the English was clear and well written.
Reviewer 3 Report
General comments:
This is a well written manuscript describing an interesting study concerned with testing a novel method for treating ileus in horses. The authors describe recovery and battery issues with some of the endoscopy capsules used in the project, which appears to have resulted in less data from this experiment than expected/intended. I am not clear on how data from (all) recovered capsules/ capsules with truncated videos were handled exactly during the analysis steps and I have asked more specific questions about this below. Similarly, while a thorough description of the study design is provided, I am not sure that I understand all aspects of how this study was set up (specific comments below). I feel like a graphic outlining the study design would be very helpful.
Specific comments:
Line 36: Typo “purpose”
Lines 41-42:
- Would it be possible to explain in more detail how many horses were in the treatment and control groups for each trial?
- Also, would it be possible to include information regarding how long the bit chewing and control conditions lasted overall before the washout?
- Could you further comment (in the main body of the paper) on how you selected 20 min for an adequate bit chewing time?
Line 56: A reference seems missing. The prevalence of ileus in patients with small intestinal lesion could be included.
Line 73 etc.:
- When the previous equine studies are described, please consider specifying whether healthy horses were used in these studies.
- Also, I noticed here, but the issue might have already happened earlier in the manuscript, that the references don’t seem to match the statements. E.g., reference 10 is indicated, but I believe reference 11 is the one that was meant. Please ensure that all references match the statements throughout the manuscript.
Line 98: In the abstract you say that 10 horses were enrolled in the main project, but here it seems like it was only 9. Please consider making it clearer how many horses were used for the main study and how many were in which group for each of the two trials/periods.
Line 122: You say ‘similarly’, what was different between trial 1 and trial 2?
Line 146: Could you comment on why 3 capsules were used? Was that based on previous studies? Did you expect the capsules to get lost?
Line 158: Could you define the end of the study period here again? Are you referring to the 2 weeks before the washout or the end of the study (after crossover etc.).
Line 159-160: Could you include a picture of how the bit was held in place?
Line 176: It seems like you used the methods of Doherty et al., 1998. Please consider including this reference in this paragraph to reflect use of their methods (sampling timepoints etc.).
Line 185: This is more of a general question about the capsule technique. I think I read that an equine stomach is contracting regularly, even if the horse is fasted. So, if the capsule is administered right after contraction vs. right before one, does that cause biases in the data for GET? Is this data reliable? Is there any research data in general on how accurate the capsule recordings are, also given that they appear to get stuck quite a bit? Is there a way of commenting on this a bit more in the discussion?
Lines 192-194: Extending the study period means that the bit chewing was continued? So not all horses had a 2 week “trial 1” period? And not all horses did trials 1 and 2, as well as the washout period at the same time? Is there a way to clarify this in the text?
Lines 194-196: I am very confused about this. I thought that horses did trial 1 (2 weeks + extension if capsules were not retrieved), washout (4 weeks), and trial 2 (2 weeks+ extension if capsules were not retrieved). Were all horses in the hospital for 1 month for trial 1? And when you say discharged, do you mean send back to their owners? I thought they went back to the pasture that they came from at the hospital/research facility. Please clarify.
Line 208: This is more of a general comment about the stats. I am very unclear on how the data from individual capsules was treated. If 1+ capsules were available per horse, was the data averaged? Was only 1 capsule per horse randomly selected for analysis, or the one with the shortest passage time? How good was the agreement for GET etc. for multiple capsules from the same horse? Also, it would be very insightful to add an agreement test for the two GET methods. Based on your table 2 data, the agreement does not seem to be very good.
Line 209-211: Given that your power test suggested n=6, and you used n=10 (or 9?), please consider commenting on the power you expected with your actual sample size used (n= 10 or 9).
Lines 233 – 235: What does that mean for the sample size of your OCTT data? If the final sample size was < n= 20/18 observations (if n= 10 or 9 per trial), please consider including the sample sizes along with all the results. This also applies to SITT data if the battery died resulting in a truncated video and causing a sample size of <20/18 videos used in analysis.
Lines 242-243: Please include the sample sizes with the CVs. How many horses, how many capsules?
Line 244: Would it be better to say that there was a reduction, but it was not sign., like you do for the other data? That would make it more consistent.
Line 249: I think your first 2 p-values do not match the table 2 p-values?
Line 253: There seems to be a superfluous “5h”, and for SITT median and mean seem switched?
Table 2: There appears to be a distinct difference between the results of the 2 GET methods. Looking at the max values, the capsule suggests 12h for a horse, while the acetaminophen method suggests <1h for that animal. It thus appears that the capsule data might be rather inaccurate for GET (we don’t know about SITT and OCTT). It may be worth re-analyzing this data using only the fastest capsule per horse, and removing outliers and lodged cameras, before running another Wilcoxon. I feel like this might result in more accurate data.
Line 309: Please consider specifying that this was an equine study.
Line 330: I thought there were 24 recovered pre-mortem +3 postmortem?
Line 375: How about GET and SITT? Should those also be studied again?
Round 2
Reviewer 2 Report
Thank you for your edits and additions to clarify the details of the project.